

# Cloning expression and immunogenicity analysis of inhibin gene in Ye Mule Aries sheep

Zengwen Huang[1,*], Juan Zhang[1,*], WuReliHazi Hazihan[2], Zhengyun Cai[1], Guosheng Xin[1], Xiaofang Feng[1] and Yaling Gu[1]

[1] Agriculture College, Ningxia University, Yinchuan, China
[2] College of Animal Science and Technology, Shihezi University, Shihezi, China
[*] These authors contributed equally to this work.

## ABSTRACT

**Background**. Ye Mule Aries sheep is one of the most important sheep breeds in Xinjiang, China. This breed is well adapted to harsh environmental conditions and displays strong disease resistance, fast growth, and high cold tolerance. To analyze the clonal expression and immunogenicity of the Ye Mule Aries sheep inhibin gene, total RNA was extracted from sheep ovarian tissue and used as a template to generate a eukaryotic expression vector and study inhibin immunogenicity.

**Methods**. Primers were designed to amplify the inhibin A gene via polymerase chain reaction and the amplified product was cloned between the *ScalI* and *EcoRI* restriction sites of the expression vector pEGFP-N1 to construct a recombinant plasmid, pEGFP-INHα. Following the validation of successful cloning, the pEGFP-INHα plasmid was transfected into BHK cells to verify expression in eukaryotes and subsequently utilized as an antigen in rabbits. Rabbits were tested for anti-inhibin antibodies and serum follicle-stimulating hormone (FSH) concentrations.

**Results**. The analysis of the INHα gene sequence revealed that INHα is 1109 bp long and is translated to an approximately 40 KDa protein. Bioinformatics approach indicated that the INHα gene is highly conserved between organisms. Immunization with the eukaryotic expression vector, pEGFP-INHα, which expresses the INHα gene elicited immune response and generatigeneration on of anti-INHα antibody. The antibody had a significant regulatory effect on the serum concentration of FSH in rabbits and led to higher levels of FSH, indicating increased ovary function.

**Conclusions**. The present work resulted in a successful construction of eukaryotic expression plasmid pEGFP-INHα and verified the immunogenicity of this highly conserved protein. Further, the expression of pEGFP-INHα was shown to have a significant impact on the secretion of FSH, indicating a potential regulatory role in ovarian function. In conclusion, our current findings can serve as a working model for studying the effect of INHα on the breeding performance of Ye Mule Aries sheep, providing a novel strategy to improve their reproduction rates.

Corresponding authors
WuReliHazi Hazihan,
1508217366@qq.com
Yaling Gu, guyaling@sina.com

## BACKGROUND

In 1932, McCullough established that a factor contained in the aqueous extract of bovine testes exerted negative feedback on follicle-stimulating hormone (FSH) secretion; the factor was named "inhibin" (INH or IB) (*Jianchen & Xiaorong, 1998*). INH is a glycoprotein hormone secreted by testicular supporting cells and ovarian granulosa cells. The functional form of INH is a dimer containing two disulfide bonds between α- and ß- subunits, as well as glycosylation sites on the base (*Medan et al., 2007*). In mammals, the ß-subunit has two isoforms, A and B. Therefore, INH is expressed in two forms: INHA (aßA) and INHB (aßB)(*Meldi, Gaconnet & Mayo, 2013*). It has been demonstrated that INH is the main negative feedback regulator of FSH secretion in mammals, suppressing the secretion of FSH in the pituitary gland (*Kaneko et al., 1995*; *Roser, Mccue & Hoye, 1994*; *Findlay, Clarke & Robertson, 1990*; *Nambo et al., 1998*; *Shi, Mochida & Suzuki, 2000*). Moreover, the levels of INH can reflect the amount of follicle growth at the beginning of the menstrual cycle, serving as a key regulator of ovulation in animals(*Kogo et al., 1993*). Additionally, INH acts locally, directly inhibiting follicular function and development of gonads (*Knight & Glister, 2001*; *Li et al., 2009a*; *Vale et al., 1988*). Several studies demonstrated that female animals immunized against INH enter the puberty earlier, show an increase in ovulation rate, and display improved fecundity (*Medan et al., 2004*; *Ishigame et al., 2004*). Both active and passive immunization with INH increase ovulation rate and lambing in sheep(*Kusina et al., 1995*; *Fray, Wrathall & Knight, 1994*; *Henderson et al., 1984*) and have a similar effect on ovulation and litter size in pigs and cattle (*Wheaton et al., 1998*; *Morris et al., 1993*). Immunization of rats with either an INH and GFP fusion gene or an inhibin gene vaccine and immunoadjuvant leads to the production of anti-inhibin antibody (*Shuilian et al., 2012*; *Dachan et al., 2003*). Moreover, the INH gene-based vaccine immunization can effectively induce an immune response in postpartum dairy cows (*Shuilian, Zhong & Wenping, 2014*). It was also shown that immunization of cows with INH enhances ovulation levels, resulting in an increased number and quality of embryos (*Li et al., 2009b*; *Chao et al., 2009*). Active INH immunization can also produce similar effects in goats (*Guiqiong et al., 2003*; *Tian & Huang, 2010*). Recent developments in molecular biology elucidated the significance of INH in regulating the reproductive function of animals. Therefore, based on studies in mice, poultry, pigs, cattle, sheep and other animals, INH immunization has the potential to improve animal fecundity, which is of great significance in both scientific research and agricultural production.

Ye Mule Aries sheep (formerly known as Kalamu Mule) is a flock of sheep located in the western margin of the Junggar Basin in Emin County, Tacheng District, Xinjiang, China(*Anivash et al., 2006*). The Ye Mule Aries sheep, characterized by a small fat hip and fat body, were created in the 19th century through the long-term breeding of Kazakh sheep in a unique geographical location with a specific climate (*Jahan & Arnivash, 2010*). Due to its desirable characteristics, the Ye Mule Aries sheep was heavily utilized in the development of the meat industry, in particular for lamb production in Xinjiang, leading to a sharp decline in the number of purebred sheep in Xinjiang. The decrease in population size has made the selection and breeding process challenging and placed the animals in

danger of extinction. To overcome this problem and design the strategy to protect Ye Mule Aries sheep population and improve their productivity, the Ye Mule Aries sheep inhibin subunit α (INH α) gene was cloned, and expression vector (pEGFP-INH α) was constructed. Next, we tested the hypothesis that eukaryotic expression of the pEGFP-INHα vector can increase ovulation rate in adult rabbits. The accumulated results demonstrated that the expression of the INHα vector significantly improved the ovulation rate in rabbits, providing a rationale for future studies involving gene vaccination.

## MATERIALS AND METHODS

**Ethics statement:** All animal care and experimental procedures were approved by the Animal Protection and Use Committee of Ningxia University and Shihezi University. All research was carried out in strict accordance with Ningxia University and Shihezi University experimental animal welfare and ethical guidelines.

**Experimental animals:** Ye Mule Aries sheep were provided by Sheep Farm, Ermin County, Xinjiang, and Japanese big white rabbits were provided by Animal Experimental Center of Shihezi University. Animals were utilized during the estrus period.

**Materials:** The cloning vector pMD18-T, DH5α bacteria, BHK cells, and eukaryotic expression plasmid pFGFP-N1 were all provided by the Oasis Ecological Laboratory of Xinjiang Production and Construction Corps of China. Lipofectamine 2000 was obtained from Invitrogen, T4 DNA ligase, 10xT4 DNA ligase buffer, and other molecular biology reagents were provided by TIANGEN, Takala, Kangwei Companies.

### Collection of Ye Mule Aries sheep ovarian tissue

The ovaries of purebred Ye Mule Aries sheep (healthy ewes that normally produce 3 fetuses) were collected from animals from the pastoral area of Halamumul Township, Emin County, Xinjiang. The ovaries were collected in 1.5 mL centrifuge tubes and placed in a gauze bag. Subsequently, the ovaries were snap-frozen in liquid nitrogen and stored at −80 °C until use.

### Extraction of total RNA and the design of synthetic primers

After thoroughly grinding the collected ovarian tissue in liquid nitrogen, total RNA was extracted according to the specification of TRNzol total RNA extraction kit (TIANGEN) and stored at −80 °C. Because eukaryotic genes contain introns, the total RNA of Ye Mule Aries sheep ovary tissue was amplified by RT-PCR to obtain the cDNA fragment of INHα subunit. Primers were designed on the basis on the full-length sequence of INHα (GenBank, XM 004.004955.1). The expected amplified product fragment size was 1,100 bp. The upstream primer, 5′-ATG TGG CTT CAG CTG CTC CTC TTC-3′, and downstream primer, 5-′GAT GCA AGC ACA GTG CTG GGT G-3′, were synthesized by Shanghai Shenggong Bioengineering Co., Ltd.

### Reverse transcription, PCR amplification, and ligation of the INHα gene and vector

Reverse transcription was carried out using a reagent kit (Takala) according to the manufacturer's instructions. The reaction mixture, 20 μL, consisted of 5×Mix (four μL),

RNA (two µg), and RNase-free water (16 µL). The reaction was performed at 37 °C for 15 min followed by 85 °C for 5 s; the product was stored at −4 °C.

PCR of the cDNA was performed according to the TaKaRa kit instructions using primers synthesized by Shanghai Biotech Co., Ltd., to obtain the target gene INHα. The reaction mixture, 25 µL, consisted of 2×Mix (12.5 µL), upstream and downstream primers (100 µmol/L, 0.4 µL each), double-distilled water (9.7 µL), and cDNA (two µL, 1.9 ng/µL). After pre-denaturation at 94 °C for 5 min, 35 cycles of 94 °C for 40 s, 68 °C for 40 s, and 72 °C for 1 min 30 s were performed, followed by a final extension at 72 °C for 7 min. The amplified product was subjected to electrophoresis on 1.0% agarose gel.

After the electrophoresis, DNA was recovered and purified according to the instructions of the QuickGel Extraction Kit (Kangwei Co.). Subsequently, the amplified sequence was ligated into the pMD18-T vector at 16 °C overnight. DH5a competent cells were transformed with the ligation product and cultured overnight on LB-coated plates containing 50 µg/mL ampicillin. Five positive clones were selected, and the identity of the recombinant plasmid was initially confirmed by PCR and double enzyme digestion.

### Sequencing of the INHα gene and pMD18-T vector

The clones were sequenced by Shanghai Shenggong Bioengineering Co., Ltd. Obtained cDNA sequences were analyzed using DNA-MAN, DNAStar, and Chromas software, as well as by the comparison of the sequence alignment with the Ara cDNA in the GenBank database (accession number XM-004004955).

### Construction and characterization of pEGFP-INHα plasmid

The pMD18-INHα plasmid and the eukaryotic expression vector pEGFP-N1 were cut at restriction sites by *ScaII* and *EcoRI* enzymes for 4 h at 37 °C. The digested products were subjected to electrophoresis on 1.0% agarose gel, and recovered and purified using a gel recovery kit. The reaction mixture for the construction of recombinant plasmid pEGFP-INHα consisted of the fragment of the pMD18 plasmid containing INHα (six µL, 1.9 ng/µL), eukaryotic expression vector pEGFP-N1 (one µL, 1.9 ng/µL), T4 DNA Ligase (one µL), 10x T4 DNA ligase buffer (two µL), and double-distilled water (10 µL), for a total volume of 20 µL. The ligation reaction was carried out overnight at 16 °C. The ligation product was transformed into *E. coli* DH5a competent cells, plated on plates containing LB medium with 50 µg/mL kanamycin, and cultured overnight at 37 °C. Five positive clones were randomly selected and the recombinant pEGFP-INHα plasmid was identified by PCR and double enzyme digestion.

### Sequencing of the pEGFP-INHα plasmid

The five pEGFP-INHα-positive clones were sequenced by Shanghai Shenggong Bioengineering Technology Service Co., Ltd. Obtained cDNA sequences were analyzed with DNA-MAN, DNAStar, and Mega 5.0 software, and aligned to the INHα subunit gene sequence in the GenBank database.

## Transfection of BHK cells and identification of pEGFP-INHα positive cells

The recombinant pEGFP-INHα plasmid was purified and recovered. The plasmid was mixed with Lipofectamine 2000 at a ratio of 4:1 and incubated with BHK cells at 37 °C in the presence of 5% $CO_2$. After 48 h, over 50% of the cells were positive for green fluorescent protein (GFP), as determined under an inverted epifluorescence microscope. Cells were harvested and divided into two groups, which were subjected to RT-PCR and Western blot analysis, respectively.

## Preparation of immunogens and immunization of rabbits

The eukaryotic expression plasmid pEGFP-INHα (optical density ratio OD260/OD280 of 1.8) was mixed with Lipofectamine 2000 to yield the final concentration of one mg/mL. The rabbits to be immunized were injected with 0.2 mL of 0.5% procaine hydrochloride into the muscles on both sides of the leg at 24 and 2 h before the immunization. Each rabbit received an injection of 0.2 mL of the plasmid (one mg/mL) at the same site. After 10 days, the rabbits were immunized again by injection at the same site, but the immunization dose was halved. Control rabbits received an equal amount of physiological saline.

## Determination of antibody levels

Antibody levels in the serum of experimental and control rabbits were determined by enzyme-linked immunosorbent assay (ELISA). For this purpose, 100 µL of a coating solution containing 15 µg of the antigen was added to the wells of a plate and incubated overnight at 4 °C. Blood samples were diluted 1:12,800 with 5% skim milk, and 100 µL aliquot was added to the coated wells and incubated at 37 °C for 1 h. After washing, 100 µL of biotinylated antibody diluted 1:1,000 was added and incubated at 37 °C for 1 h. Wells were washed again and 100 µL of streptavidin-horseradish peroxidase (HRP, diluted 1:1,000) was added and incubated at 37 °C for 1 h. After the final washing, color development was carried out by TMB (Sigma Biochemical Co., Ltd.), and the absorbance value was read at 450 nm.

## Determination of FSH hormone levels

The levels of FSH in serum obtained from the experimental and control rabbits was determined by radioimmunoassay. The assay kit was purchased from the Northern Institute of Biotechnology. The range of the standard curve was adjusted to be consistent with the range of normal animal hormone levels, as recommended by the manufacturer.

## Statistical analysis

Results are expressed as mean ± standard deviation. Data were analyzed by one-way analysis of variance (ANOVA) with Duncan's Multiple Range test used for pairwise comparisons. All calculations were performed with the SAS 9.2 software (SAS Inst., Cary, North Carolina, USA). Values were considered significantly different at $P < 0.05$.

## Availability of data and materials

The raw data has been submitted to the National Center for Biotechnology Information (NCBI) Sequence Read Archive (SRA), and the accession number is KP-113678.1.

## RESULTS

### Cloning of the Ye Mule Aries sheep INHα gene coding region and construction of recombinant plasmid

Total RNA extracted from the ovary tissue of Ye Mule Aries sheep (absorbance of A260/A280, 1.9) was subjected to agarose gel electrophoresis. Two bands corresponding to the 18S and 28S rRNA were clearly visible (Fig. 1A), indicating that no degradation occurred in the extraction process. Total RNA from the ovary tissue of Ye Mule Aries sheep was reverse transcribed and amplified by PCR using Oligo dT primers. The obtained product was subjected to agarose gel electrophoresis and yielded a specific band corresponding to the length of 1,109 bp (Fig. 1B), which was preliminarily identified as the Ye Mule Aries sheep INH α subunit coding region. Subsequently, the product was incorporated into the PMD18-T vector and transformed into E. coli DH5a competent cells. The identity of the cloned gene (KP-113678.1) was confirmed by PCR, restriction enzyme digestion, and sequencing.

In order to express INHα in eukaryotic organisms, the recovered and purified INHα subunit gene was ligated with a linearized eukaryotic expression vector, pEGFP-N1. The ligation product was transfected into E. coli DH5a competent cells, which were cultured in the presence of kanamycin. The formed colonies were verified by PCR and restriction enzyme digestion (Fig. 1C). Moreover, positive colonies were sequenced (Shanghai Shenggong Biological Engineering Technology Service co., LTD); the obtained sequence was consistent with that of the cloned gene, indicating that the pEGFP-INHα eukaryotic expression plasmid was successfully constructed.

### Molecular biology analysis of Ye Mule Aries sheep INHα

The cloned expression vector pMD18-INHα was sequenced and detected by Shanghai Shenggong Bioengineering Co., Ltd. The results showed that the full length of the inhibin gene was 1109 bp, in accordance with the target band length, of which 1083 bp constituted the open reading frame (ORF). The ORF consists of the start codon ATG, stop codon TAA, and contains the complete INH α subunit coding region, which codes for 360 amino acid residues. The A, T, G, and C content in the cDNA sequence of the INHα gene was analyzed using DNAMAN software. The percentages of each nucleotide were 13.6%, 20.3%, 30.0%, and 36.1%, respectively. The total G+C content (66.1%) was higher than that of A+T (33.9%). The calculated molecular weight of the protein was 39729 Da. Sequence comparison with the GenBank *Ovis aries* sequence (accession number: XM_004004955.1) revealed two point mutations (G580A, A636G), giving a nucleotide sequence homology of 96.94% and amino acid homology of 98.90%. Moreover, the nucleotide sequence of the peptide moiety was the same. Additionally, SignalP4.1 software (http://www.cbs.dtu.dk/services/SignalP/) analysis predicted that there was a signal peptide cut point between the 17th and 18th amino acids of the INHα protein ($D = 0.861$, D-cutoff = 0.450). Thus, the entire protein includes a 17 amino acid signal peptide and a 343 amino acid mature peptide (Figure2A). Tthe NCBI database (http://www.ncbi.nlm.nih.gov/Structure/cdd/wrpsb.cgi) on-line tool Conserved Domain Search Service identified the presence of a transforming growth factor (TGF)-beta domain

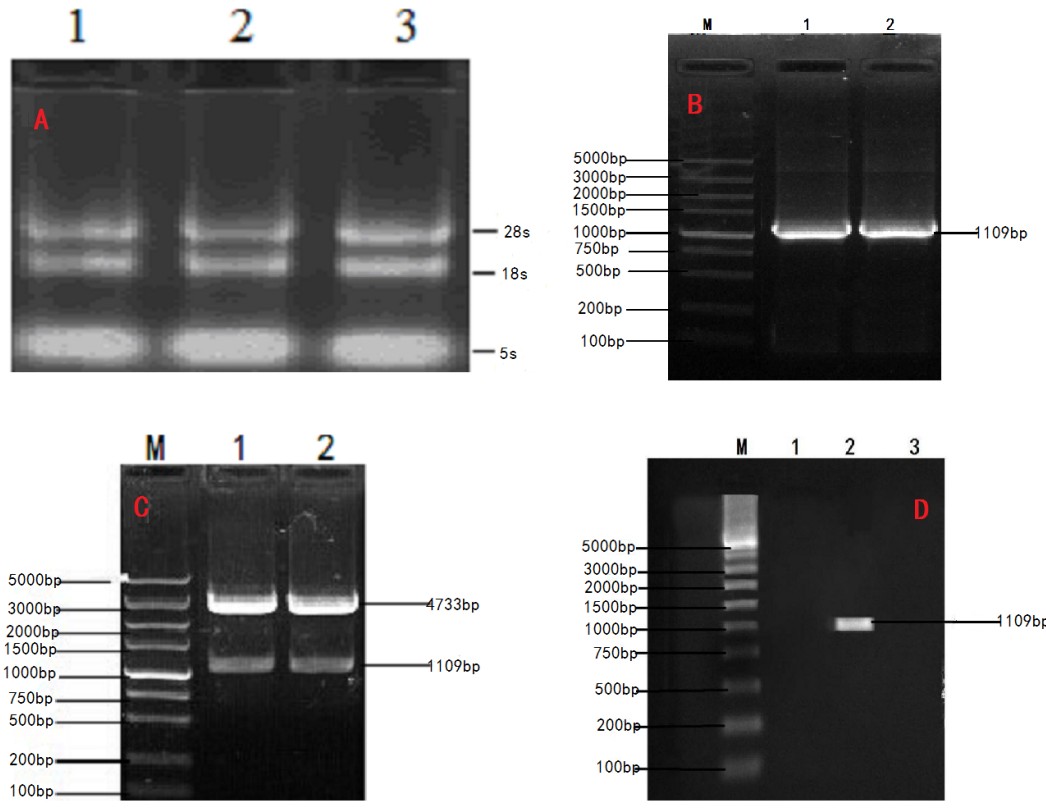

**Figure 1  Cloning and verification of objective Gene INHα.** (A) Agarose electrophoresis of total RNA;1.2.3. The total RNA were Ye Mule Aries sheep ovarian tissue. Two ribosomal RNA bands of 18S and 28S were clearly visible indicating that the RNA of Ye Mule Aries sheep ovarian tissue no degradation occurred in the extraction process. (B) PCR product of INHα subunit gene in Ye Mule Aries sheep; M, DL5000 DNA Marker; 1and 2 The amplification product of INHα gene. (C) Enzyme digestion identification of recombinant plasmid PEGFP-INHα; M,DL5000 DNA Marker; 1 and 2 pEGFP-INHα double enzyme digestion products. The eukaryotic expression vector pEGFP-INHα was verified by PCR and enzyme digestion. (D) Identification of recombinant plasmid expression in cells; M: DL5000 DNA Marker; 1. pEGFP-N1 in BHK cells expressed; 2. pEGFP-INHα in BHK cells; 3. No BHK cell transfection expressed. RT-PCR proved that the constructed vector was successfully transfected into BHK cells.

of at the site from amino acid 253 to 360 of the INHα protein. Moreover, the growth factor TGF-beta domain and one transforming growth factor TGF-ßfamily member active domain were identified at amino acids 256 to 360 (Fig. 2B).

Results obtained using DNAMAN software showed that the length of the amplified INHα gene was 1109bp and the homology to the sequence of human and gorilla genes was 84.4% and 84.3%, respectively. The homology to cattle, horse, goat, sheep, pig, wild boar, dog, cat, rabbit, *Mus musculus*, brown rat, and rainbow trout genes was 95.2%, 86.1%, 97.9%, 99.8%, 88.1%, 88.4%, 83.6%, 85.5%, 83.4%, 80.4%, 79.0%, and 51.5%, respectively. The comparison of the Ye Mule Aries sheep follicle inhibin gene with other animals revealed that Ye Mule Aries sheep gene had the highest homology with sheep, followed by goat, cattle, wild boar, pig, horse, cat, human, large orangutan, dog, rabbit, *Mus musculus*, brown rat and rainbow trout genes. This data shows that the INHα gene has

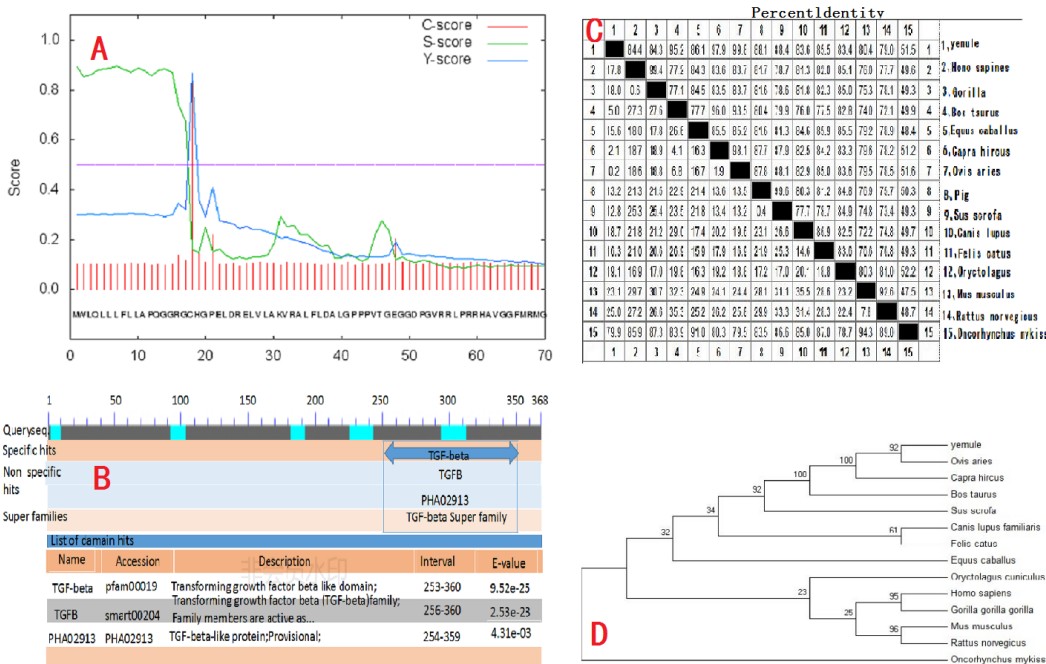

**Figure 2 Molecular biology analysis of Ye Mule Aries sheep INHα.** (A) By signalP4.1 software was used to predict signal peptides of Ye MuLe Aries sheep INHα gene. (B) Structural domain of INHα protein analyse, identified Ye Mule Aries sheep growth factor TGF-beta domain and one transforming growth factor TGF-ßfamily member active domain are present. (C) Ye Mule Aries sheep INHα Sequence alignment analyse with other species, these data represents that the follistatin gene is highly conservative. (D) Ye mule Aries sheep INHα gene DNA sequences of genetic evolution analyse, by evolutionary tree, we can find that the variation of follistatin gene of Ye Mule Aries sheep is also in accordance with the natural law of animal evolution.

relatively high homology between different animals, which may indicate that the ability of INHα immunization to increase fecundity is preserved across species (Fig. 2C).

The phylogenetic tree analysis of the Ye Mule Aries sheep INHα gene by Mega5.0 software indicated that this gene follows the expected evolutionary genetics, from aquatic to terrestrial animals, and from low to high organisms. Based on the alignment of the homologues of the INHα subunit gene between the Ye Mule Aries sheep and other animals, the phylogenetic tree clearly reflects the evolutionary genetic characteristics of the organism (Fig. 2D).

## Identification of recombinant plasmids in BHK cells by RT-PCR and western blot

The transfection efficiency of the recombinant plasmid (pEGFP-INH α) in BHK cells was analyzed 48 h after transfection on the basis of green fluorescence (Figs. 3A–3C). Moreover, a specific band of 1,109 bp was detected in BHK cells by RT-PCR, which was absent in non-transfected BHK cells (Fig. 1D). The expression of the recombinant plasmid pEGFP-INHα was also confirmed by detection of a specific 40 kDa band in western blot

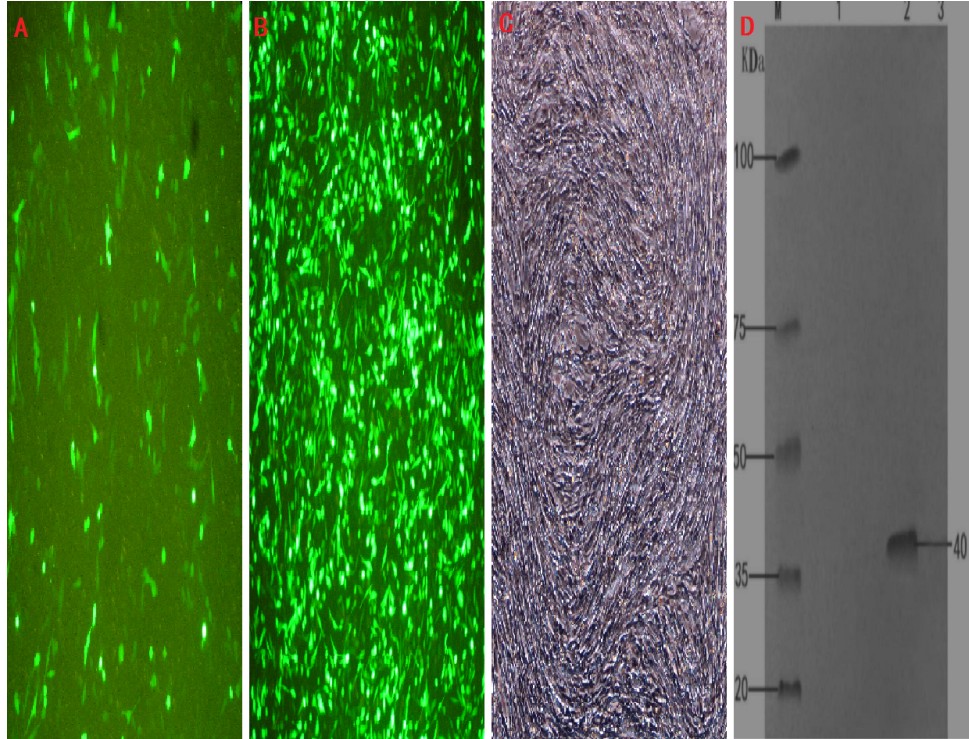

**Figure 3   Cell infection experimental analysis and verification of the target gene INHα.** (A) pEGFP-N1 in BHK cells expressed; (B) pEGFP-INHα in BHK cells expressed; (C) No BHK cell transfection. (D) Western blotting to identify protein expression in recombinant plasmids.M. proteinMaker; 1.pEGFP-N1 in BHK cells expressed; 2. pEGFP-INHα in BHK cells; 3. No BHK cell transfection expressed.

assay. Thus, the pEGFP-INHα plasmid was successfully constructed and expressed in eukaryotic cells (Fig. 3D).

## Antibody titer in blood

The level of anti-inhibin antibody in the blood of the immunized rabbits, measured by optical density (OD), gradually increased from day 10 after the first immunization. The initial immunization was followed by the booster immunization after 10 days. All OD measurements were significantly higher in the injected animals than in the control group. These results indicate that antibody production was rapidly stimulated in the rabbits by the primary immunization, and the antibody level in the blood continued to rise after the second immunization, while anti-inhibin antibody was not produced in the control group (Table 1).

## Plasma FSH level

After the initial immunization, the levels of FSH in the serum of the immunized group were slightly higher than in the control group. Ten days after the first immunization and 10 days after the second booster immunization, the serum FSH concentration in the immunized rabbits was significantly higher than in the controls. This result indicates that the first

**Table 1  Changes in inhibin antibody titer in rabbit blood.**

| Processing group | Antibody titer | | |
|---|---|---|---|
| | 0d | 10d | 20d |
| Immunization group | $0.09 \pm 0.002$ | $0.75 \pm 0.147^{A}$ | $1.24 \pm 0.123^{A}$ |
| Control group | $0.09 \pm 0.001$ | $0.09 \pm 0.169^{B}$ | $0.11 \pm 0.030^{B}$ |

Notes.
For data in the same column, different lowercase letters indicate significant differences ($P < 0.05$), while different uppercase letters indicate extremely significant differences ($P < 0.01$).

**Table 2  Effect of immunosuppressive factor on serum FSH mIU/mL.**

| Processing group | Serum FSH concentration | | |
|---|---|---|---|
| | 0d | 10d | 20d |
| Immunization group | $1.32 \pm 0.45$ | $1.68 \pm 0.38$ | $2.68 \pm 0.34^{A}$ |
| Control group | $1.40 \pm 0.40$ | $1.39 \pm 0.22$ | $1.35 \pm 0.16^{B}$ |

Notes.
Uppercase English letters indicate that the difference is extremely significant ($p < 0.01$), and the addition and subtraction after the average is expressed as SE.

immunization with INHα caused a slight increase in serum levels of FSH, which were further significantly increased following the booster immunization (Table 2).

# DISCUSSION

Previously, the mechanisms regulating reproductive traits in animals have been examined in the sheep (*Forage et al., 1986*), cattle(*Rodgers, 1991*), pigs (*Mason et al., 1985*), humans (*Mason, Niafl & Seeburg, 1986*), and mice (*Woodruff, Meunier & Jones, 1987*). The present study represents the first effort to clone the INHα subunit gene of the Ye Mule Aries sheep. The gene has a coding region of 1,109 bp, and the complete open reading frame of this sequence contains 1,083 nucleotides encoding 360 amino acids. The first 17 amino acids are predicted to function as a signal peptide and amino acids 18 to 360 form the remainder of the mature INHα protein. The domain analysis of INHα protein showed that this protein contains one TGF family domain and one TGF precursor superfamily domain, and has the activity of autocrine and paracrine growth factors. It binds to the TGF family receptors and TGF superfamily receptors and participates in the growth and development of the pituitary, gonads, placenta and other organs (*Ying, 1988*; *Vale & Rivier, 1986*; *Findlay, 1993*). By conducting a comparison analysis, we showed that the homology of the nucleotide and amino acid sequences of the Ye Mule Aries sheep, common sheep, goat, and domestic cattle were over 95%; this finding was supported by the functional bioinformatics analysis of the INH α protein of the Ye Mule Aries sheep.

In this study, we have successfully constructed the eukaryotic expression plasmid pEGFP-INHα and expressed it in BHK cells, along with the eukaryotic expression vector pEGFP-N1 coding for the enhanced green fluorescent protein (EGFP), which was used to detect the expression of exogenous genes *in vitro*. The expression vector is more convenient for detecting the expression of the fusion gene (*Chen et al., 2002*). The eukaryotic expression system developed here overcomes certain deficiencies present in prokaryotic expression

systems because the expression product retains the natural activity of the original protein. Moreover, the expression product is non-toxic and easy to purify, making this system increasingly attractive (*Jixian et al., 2011*).

Previous work has shown that fertility-regulating inhibitors can bind to activin, thus effectively preventing morphological changes in granulosa cells induced by activin, resulting in specific inhibition of FSH release and the maintenance of normal ovarian function. Proper control of FSH is vital due to its effects on follicular growth, development, luteinization, and regulation. Thus, the function of granulosa cells ultimately determines the luteinization and atresia of follicles (*Findlay, 1993*). Gene immunization is based on the same basic principles as standard immunologic procedures, and inhibin gene immunization is based on conventional gene immunization and INH (*Jiang, 2004*). Therefore, the inhibin antigen-encoding gene can be inserted into a eukaryotic vector and transformed into the animal, leading to the synthesis of the inhibin antigen protein by the transcriptional system of host cells and its secretion. These processes trigger the generation of the specific immune response of the host and production of an anti-inhibin antibody to neutralize the follicle inhibin. The resulting interaction between the proteins reduces the levels of inhibin, thereby increasing the ovulation rate and sperm production in animals (*Yong, Min & Wenju, 2003*). Given the regulatory mechanism of the inhibin gene and the negative feedback regulation of FSH, as well as current genetic vaccine methods, such as the imatin DNA vaccine pINH immunization of mice (*Xunping et al., 2002*), eukaryotic expression plasmid pcINH active immunization of rats (*Mao et al., 2003*), pCIS recombinant plasmid active immunization of rats (*Dachan et al., 2003*), and other immunization methods, the results obtained here indicate that inhibin gene immunity can promote follicular development and increase plasma FSH levels.

In this study, a pEGFP-INHα plasmid was constructed and adopted a scientific immunization method to actively immunize rabbits using the corresponding adjuvant. Anti-inhibin antibodies could be detected in the blood after the first immunization. However the titer of antibodies remained low until the second immunization, which increased antibodies production and resulted in higher titer of the egg yolk antibody (*Zhang et al., 2010*). Our finding provides a basis for further investigation of genetically engineered inhibin vaccines, especially using genetic recombination technology which has greatly simplified this process. Nevertheless, for practical applications, some macromolecular proteins will require specific structure analysis and selection of suitable vectors and strains to ensure correct expression and preserve the structural and physical properties of the Inhibin A subunits (*Yan et al., 2003*). This strategy opens a novel avenue for the establishment of more specific monoclonal antibodies and has great potential as a new tool for improving the overall fecundity of animals.

Importantly, increasing the fecundity of sheep through immunization has unique advantages, such as high efficiency, stability, simplicity of operation, and broad applications, and thus deserves widespread research attention. A large number of experiments on merino sheep and the merlin merino hybrid ewes in the border area have confirmed that active inhibin immunization can increase the average ovulation of each ewe from 1.2 to 4.0 (*Forage et al., 1987*). In addition, several experimental approaches proved that inhibin

immunization not only increases the rate of ovulation but also increases the number of lambs (*Wheaton, Carlson & Kusina, 1992*; *Glencross et al., 1994*), as well as ewes (*Wrathall et al., 1990*), mice (*Sewani et al., 1998*) and rats(*Yin, Xuting & Xiaodon, 1999*). Inhibin gene immunization can theoretically reduce the level of inhibin *in vivo* for an extended time, and its immune effect should be equivalent to or better than conventional immunity (*Forage et al., 1987*; *Glencross et al., 1994*; *Wrathall et al., 1990*).

Inhibin gene immunization can lower the levels of inhibin over a long time. The immune effect should be comparable to, or even better than, the effect of routine immunization. As a result, the technology offers the advantages of high stability, simple operation, and easy production. It can be used in correcting past genetic selection, embryo transfer, superovulation, and hormone-induced twins technology. Thus, inhibin gene immunization became in recent years the focus of intense research aiming at increasing the lambing rate of sheep. Further research on the mechanism of action and physiological response pathways of the inhibin gene will, most likely, benefit the rapid, healthy, and sustainable development of the sheep industry, and solve the "lambing rate" problem of Xinjiang sheep in the future.

## CONCLUSION

The present study showed that the Ye Mule Aries sheep follicle inhibin gene was successfully integrated into a highly efficient and stable INHα eukaryotic expression system yielding a biologically active protein. Moreover, the inhibin gene is a regulator of genes responsible for reproductive function and can be utilized to improve the fertility and production performance of animals via immunization-based reduction of the negative feedback of inhibin on the follicle stimulating hormone, estrogen.

## ACKNOWLEDGEMENTS

The authors thank all animal keepers and farmers who have provided research material for the experiments. They are also grateful to the scientific research teams of Ningxia University and Shihezi University for their concern, help, and support.

### Funding

This study was financially supported by the Science and Technology Research and Development Program of Shihezi University (gxjs2011-yz09). The funders had no role in study design, data collection and analysis, decision to publish, or preparation of the manuscript.

### Grant Disclosures

The following grant information was disclosed by the authors:
Shihezi University: gxjs2011-yz09.

### Competing Interests

The authors declare there are no competing interests.
## Author Contributions

- Zengwen Huang conceived and designed the experiments, performed the experiments, contributed reagents/materials/analysis tools, prepared figures and/or tables, authored or reviewed drafts of the paper, approved the final draft.
- Juan Zhang analyzed the data, contributed reagents/materials/analysis tools, prepared figures and/or tables, approved the final draft.
- WuReliHazi Hazihan conceived and designed the experiments, analyzed the data, contributed reagents/materials/analysis tools, prepared figures and/or tables, authored or reviewed drafts of the paper, approved the final draft.
- Zhengyun Cai analyzed the data, contributed reagents/materials/analysis tools, approved the final draft.
- Guosheng Xin prepared figures and/or tables, authored or reviewed drafts of the paper, approved the final draft.
- Xiaofang Feng performed the experiments, analyzed the data, approved the final draft.
- Yaling Gu contributed reagents/materials/analysis tools, prepared figures and/or tables, authored or reviewed drafts of the paper, approved the final draft.

## Animal Ethics

The following information was supplied relating to ethical approvals (i.e., approving body and any reference numbers):

All animal care and experimental procedures were approved by the Animal Protection and Use Committee of Ningxia University and Shihezi University (NO). All research work is carried out in strict accordance with Ningxia University and Shihezi University (SAU) experimental animal welfare and ethical guidelines.

## Data Availability

Data is available at GenBank under accession number KP113678.1 https://www.ncbi.nlm.nih.gov/nuccore/KP113678.1/.

## Supplemental Information

Supplemental information for this article can be found online at http://dx.doi.org/10.7717/peerj.7761#supplemental-information.

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
