# Peer review of "Cloning expression and immunogenicity analysis of inhibin gene in Ye Mule Aries sheep"

_PeerJ, doi:10.7717/peerj.7761_

## Round 0.1 · original submission · Major Revisions

· Academic Editor

Major Revisions

Both reviewers have raised several concerned related to the language of this manuscript which is riddled with grammatical and syntax errors. Authors should address all the concerned raised, specifically of Reviewer 1.

Reviewer 1 ·

Basic reporting

The manuscript entitled "Cloning expression and immunogenicity analysis of Inhibin Gene in Ye Mule Aries sheep" by Zengwen Huang et al. characterized the inhibin-a subunit gene (INHa) of Ye Mule Aries sheep, which is found locally in Emin County, Tacheng District, Xinjiang, China. Homologues of INHa gene have also been investigated in many other mammals in correlation to the breeding performance.

I have found serious issues with this manuscript thus I do not recommend it for publication at its present form unless the manuscript is extensively revised.

The English of the manuscript is rather poor. Not only dozens (!) of spaces are missing and plenty of typos show up everywhere, but also many times it is not easy to find out the meaning of the sentences. Most of these mistakes can easily be corrected using a spell check of text editor software, but help of an academic expert with good language skills is inevitable.

Experimental design

Experimental design is satisfactory, however the manuscript is poorly written, due to this the major impact of this manuscript is hidden.
I found many formatting mistakes like space between two words, font style and font size, subheadings of the different section. Introduction, results, and discussion sections are poorly written.

Validity of the findings

Manuscript by Zengwen Huang et al. characterized the inhibin-a subunit gene (INHa) of Ye Mule Aries sheep, which is found locally in Emin County, Tacheng District, Xinjiang, China. The study is very specific to animal of particular region and has limited global commercial values.

Additional comments

Line 11-13: Long and incomplete sentence.
Line 13-15: It is advisable to write the background, result and discussion section mostly in past tense. Sentence is ambiguous, unclear and incomplete.

Line 17-18: “The coding region sequence (GenBank accession number: XM-004004955.1) was designed to synthesize primers, and the target sequence was obtained by reverse transcription amplification.”
The sentence is technically incorrect. Coding region sequence can’t be designed. It was the primer, which was designed using coding sequence as template.

Line 18: “transcription amplification” both is verb and can’t be used in tandem at the end of the sentence.

Line 19: “cleavage sites” it should be ‘restriction sites’.

Line 22: In the sentence “The plasmid (pEGFP-INHα) was used as antigen”. Is it the plasmid that has been used as an antigen??

Line 25: “was also 1109 bp long” what else was also 1109bp??
“Protein size was about 40 KD” it is not size, proteins are depicted in molecular weight. KD can be written as KDa.

Line 29: “This study successfully constructed eukaryotic expression plasmid PEGFP-INH α and verify the plasmid PEGFP-INHα immunogenicity, through the biological software analysis and PEGFP-INHα cells and immune experiment, we get the conclusion is highly conservative INHα gene sequences, INHα gene carrier by high expression in cells, INH alpha gene within the animal body on the secretion of FSH has important influence”
Change “PEGFP-INHα” to “pEGFP-INHα”.
The biological software can predict only, and not verify the immunogenicity.
The sentence is long, not clear and abruptly ended.

Line 38-42: Since 1932, McCullagh has found a water-containing factor in the aqueous extract of bovine testes, which has a feedback-inhibiting activity on FSH secretion, named "inhibin" (Inhibin INH or IB), but scientists A large number of studies have begun on statins, but it has not been possible to obtain breakthroughs in the study of statins until Robertson et al. and Ling et al. obtained isolated and purified statins from follicular fluids of bovine and porcine, respectively [1].
Authors are advised to avoid long sentences. This sentence is very long, poorly written and full of grammatical errors.

Line 50: change “Prove” to prove.
Line 68: In to in.
Line 67 to 70 is not clear:
It is from the middle of the 19th century that the majority of Kazakh herders In the unique geographical and climatic conditions of the locality, the mutton for obtaining a large amount of Ye Mule Aries sheep and small fat hip fat body was formed from Kazakh sheep through long-term breeding.
Line 70: “It should be paid attention to in the” use of ‘to’, ‘in’, ‘the’ is grammatically incorrect.

Line 74-77: “In order to implement the policy of protecting the valuable resources of local resources and improve the productivity of the Ye Mule Aries, this experiment used Xinjiang Ye Mule Aries as the research object, cloned its inhibin a subunit gene (INHa), and constructed the gene.”
What authors want to say in the above sentence? What is the meaning of “valuable resources of local resources” in line 74-75??
“Ye Mule Aries” should be italicized throughout the manuscript.

Line 94-95: The liquid nitrogen was quickly frozen, and then placed in -80℃ refrigerator for use.
What do you mean by “liquid nitrogen was quickly frozen??

Line 100: “The amplified product fragment size is expected to be approximately 1100 bp”. “size was”.

Line 105-107: “PCR amplification of the cDNA sequence obtained by reverse transcription using the designed primers, the amplification system is 25μL, of which 2xMix 12.5μL, the upstream and downstream primers (100μmol/L) are each 0.4μL, the cDNA is 2μL, and the deionized water is added.”
Sentence should be in past tense, poorly written.

Paragraph from Line 121-130 should be merged together.

Line 132: “The cloned bacteria that were initially identified as positive were DNA-sequenced by Shanghai”
Is it the bacteria were sequenced??

Line 135: Sequence alignment analysis of the INHα subunit gene.
What do you want to say in this sentence??

Rewrite ‘Materials and Methods’ again as I give up in reviewing the section. The whole section is full of grammatical errors and so it is difficult to understand the sentences and message. I would request the authors to take the help of native English speaker.

Line181: “visible (Figure 1), indicating that no degradation occurred in the extraction process of.”
Extraction process of what??? Sentence is incomplete.

Result and Discussion section are also full of errors and authors are advised to rewrite the whole section.

Figures and figure legends needs an extensive rearrangement. Protein domain architecture in figure 4 is not acceptable! This is the screenshot of NCBI CD search. Authors are required to prepare this figure in ‘line and graph’ format of the same in any image tool like Powerpoint or Illustrator.
All figures should be grouped in 2-3 figures.

Reviewer 2 ·

Basic reporting

The manuscript titled “Cloning expression and immunogenicity analysis of 1 Inhibin Gene in Ye Mule Aries sheep (#32706)” is publishable In Peer J, but I have few concerns mentioned below:
Overall the english language should be thoroughly checked throughout the manuscript to make it more understandable.
Abstract:
In abstract, line 6, modify this sentence
“which is used as a template and according to the Aries INHα gene “
Modify this sentence
“The plasmid (pEGFP-INHα) was used as antigen and immunoassay”
Please modify english language in abstract.
Background
I will suggest to reorganize the introduction part and fix grammatical errors.
Line 39: Elaborate FSH here.
Line 39-40: Modify the sentence below
“but scientists A large number of studies have begun on statins “
Line 50: replace Prove by prove
Line 57: replave Vaccine by vaccine.
Line 68: replace In by in
Line 92: How many Ye Mule Aries sheep ovarian tissues were collected? Please define.
Line 138: What ratio?
“recovered, mixed with a transfection reagent (Lipofectamine 2000) in a certain ratio”
Line 102: Remove the primer
Line 142: Correct this sentence
“divided into two, one for RT-PCR amplification, and the other for RT-PCR amplification”

Experimental design

The experimental design is satisfactory.

Validity of the findings

No Comments

---

## Round 0.2 · Major Revisions

· Academic Editor

Major Revisions

In spite of first revision authors have failed to improve the quality of this manuscript. As pointed out by the reviewer comments, for example:

1- results lane 21- "INHα is also 1109bp long" is not acceptable.
2- Multiple abrupt ending statements; e.g: Methods lane 16: "according to the Aries INHα gene"

These are not the only examples. Overall, the manuscript is still in a bad shape and doesn't read well. It has to go through a major revision otherwise it will be rejected.

Reviewer 1 ·

Basic reporting

There are still serious issues with this revised manuscript, thus I do not recommend it for publication.

First, the study of the Ye Mule Aries sheep, which is a local breed and found in very specific geographical location, so this study doesn’t have broad significance.

Starting from the abstract, the tense of the sentences are not consistence throughout. Abstract should present the broad picture of the study and not the facsimile of the manuscript.

In Background section of the abstract, last sentence in line 17 is ending abruptly and like that there are many more grammatical and language error throughout the manuscript that I had given up.

In the method section of abstract, rather than protocol detail, author should present the broad methodology of the work. Sentences are still abrupt and transition between sentences and paragraph is still missing. It is very hard for reader to understand the manuscript.

Line 85: “INHα is also 1109bp long” author should specify what else is 1109 bp?? Why use “also” ?? This point was also raised in earlier review but not addressed by author! Like this, many more issues hadn't been addressed by the author.

Subheadings, as it was also pointed in earlier review, in methods and result section still didn’t improved. e.g. in result section the subheading “Total RNA isolation” doesn’t make sense?? What result you got from RNA isolation?? ‘RNA isolation’ is no longer mentioned as a separate subheading even in methods section and here discussed in result. All the subheadings in the result section are poorly constructed and no significant improvement is found in revised manuscript. Thus I do not recommend for publication.

Experimental design

Has been reviewed and mentioned earlier.

Validity of the findings

Not significant for broad audience.

---

## Round 0.3 · Major Revisions

· Academic Editor

Major Revisions

Please go through all the comments by the reviewer and incorporate all the changes in the revised manuscript,

Reviewer 1 ·

Basic reporting

Revised version of the manuscript entitled "Cloning expression and immunogenicity analysis of Inhibin Gene in Ye Mule Aries sheep" by Zengwen Huang et al. characterized the inhibin-a subunit gene (INHa) of Ye Mule Aries sheep.

Manuscript has significantly improved after revision but still the manuscript is full of typos, which cannot be overlooked and hence needed a major revision.

Line 12: ‘Background: Ye mule aries is one of the most important sheep breeds in Xinjiang, China, which as it is’ The use of ‘which as it is’ is incorrect, either use “as it is” or “which is”.

Line 14-16: ‘In order to analyze the clonal expression and immunogenicity of the Ye mule aries sheep inhibin follistatin gene, we extracted total RNA from sheep ovarian tissue which was used as a template to generate an eukaryotic expression vector and study its immunogenicity.’

This sentence is not clear, and it should be the part of method section of abstract.
Probably author want to state in this sentence is that ‘isolated total RNA was used as to synthesize cDNA, then this cDNA was used as a template to PCR amplify the gene which was cloned in eukaryotic expression vector.’

Line 38: (INH or IB) [1].INH: space missing after sentence.

Line94: -80℃for reserve: space missing. Please check throughout the manuscript for this type of error.

Line 84: China,Lipofectamine: space missing
company,T4: Space missing.

Generally space is provided between numerals and unit for e.g. 100μmol/L should be 100 μmol/L .

Line115: Five positive clones werer and omly picked: Please check for typos.

Line 121: “Construction and identification of pEGFP-INHα plasmid”: change identification to characterization

Line 174: Total RNA isolation: Subheading of results section is not appropriate. I had already mentioned in earlier comments. Please update accordingly.

Either bioinformatic analysis should be presented first or in the last.

Combine figure 3 to 6 into one figure.

Please combine all molecular biology work in one section and one figure.

Rewrite Acknowledgement in formal language.

Experimental design

Already provided in earlier comments.

Validity of the findings

Already provided in earlier comments.

Additional comments

Typos are present throughout the manuscript, so it is requested to re-analyze the manuscript again.

---

## Round 0.4 · Major Revisions

· Academic Editor

Major Revisions

Authors have not addressed all the questions raised by the Reviewer 1. Please address all the comments given by the reviewer and resubmit a revised manuscript.

Reviewer 1 ·

Basic reporting

Manuscript has significantly improved, and authors did satisfactory job in improving the sentences and paragraph structure, however the result section is still heavily unorganized. From the middle of nowhere, bioinformatic analysis pops up in the molecular biology section. And again RT-PCR analysis is mentioned in between protein localization and western blot data. I already commented earlier and again mention below the steps to be taken to improve the structure of result section. Hence, author need to structure the result section again and a major revision is recommended for the manuscript.

Experimental design

NA

Validity of the findings

NA

Additional comments

It will be nice to present bioinformatic analysis at the start of result section as it paves the way for further gene/protein experimental analysis. As, I also suggested in my last remark, that please move all the bioinformatic analysis of INHα gene and protein sequence as Figure 1 and accordingly move the result subheading “178 Bioinformatics analysis of Ye Mule Aries sheep INHα” as the first subheading in result section and accordingly update the text and figure number in revised result section.

As I already suggested in my last comment, please combine all DNA and RNA molecular biology work in one section. It is no longer in trend that one RNA isolation gel image or PCR gel image etc. qualifies for one single figure. Please combine figure 1, 2, 4 and 6 as figure 2 and accordingly update the manuscript as suggested above for earlier section. Please check the text and labelling in figures as text in figure 2 is not readable.

It will be nice if the protein expression and analysis work is presented as third section and figure 5 and 7 can be combined together as single figure 3.
Please improve the image quality of western blot in figure 7. Resize the image in figure 5.

Text in table 1 also needed to be improved.

Change the last sentence of ‘Acknowledgement’ as it is not required to acknowledge editor and reviewer.

---

## Round 0.5 · Minor Revisions

· Academic Editor

Minor Revisions

Please go through all the remaining (minor) changes suggested by the reviewer and submit it again.

Reviewer 1 ·

Basic reporting

Line 168: Cloning of the Ye Mule Aries sheep INHα gene coding region and Construction of Recombinant
Plasmid
Author should use uppercase/lowercase uniformly throughout the headings. Here no need to capitalize “Construction of Recombinant Plasmid”.

Line 174: The obtained produces a specific target band’
What obtained???

Line 175-176: Subsequently, the was cloned into the PMD18-INHαvector and transformed into E. coli DH5a competent cells.
What ‘the’ was cloned??

Line 186: The cloned expression vector pMD18-INHα was sequenced and detected by Shanghai Shenggong Bioengineering Co., Ltd.
What do you mean by ‘and detected by Shanghai Shenggong Bioengineering Co., Ltd.”???
The construct was sequenced by Shanghai Shenggong Bioengineering Co., Ltd. But why it is detected here???

Line 191: nucleotide was 13.6%
The sentence is plural. Use ‘were’ instead of ‘was’.

Experimental design

Alreadry stated.

Validity of the findings

Valid findings

---

## Round 0.6 · Minor Revisions

· Academic Editor

Minor Revisions

Article still have minor changes to be performed. Please address them one by one and resubmit the manuscript.

Reviewer 1 ·

Basic reporting

Still language is not great but acceptable. Authors fail to make recommended correction again. It is not possible to accept the manuscript without correcting the errors as mentioned in the last report and again in this report.

Please go through these lines carefully again and make the necessary corrections.

Line 174 The obtained produces a specific target band with the length of 1109 bp (Figure1B).


175Subsequently, the was connected to the PMD18-Tvector and transformed into E. coli DH5a competent cells

Experimental design

NA

Validity of the findings

NA

---

## Round 0.7 · accepted · Accept

· Academic Editor

Accept

Authors have addressed all the questions raised by the reviewer.

Reviewer 1 ·

Basic reporting

Manuscript has significantly improved after revision.

Experimental design

Experiments were well defined and adhered to the objective of the paper.

Validity of the findings

The findings validate the major questions raised in the manuscript.

Additional comments

Manuscript accepted.